# Geometric Characteristics of BOF Slag Coarse Aggregate and its Influence on Asphalt Concrete

**DOI:** 10.3390/ma12050741

**Published:** 2019-03-04

**Authors:** Dezhi Kong, Meizhu Chen, Jun Xie, Meiling Zhao, Chao Yang

**Affiliations:** 1State Key Laboratory of Silicate Materials for Architectures, Wuhan University of Technology, Wuhan 430070, China; kongdz@whut.edu.cn (D.K.); chenmzh@whut.edu.cn (M.C.); hbyangc@whut.edu.cn (C.Y.); 2Research Institute of Highway of Ministry of Transport, Beijing 100088, China; 197791@whut.edu.cn (M.Z.)

**Keywords:** BOF slag, coarse aggregate, geometric characteristics, asphalt concrete, road performance

## Abstract

In order to examine the geometric characteristics of BOF (blast oxygen furnace) slag coarse aggregate, the aggregate image measurement system (AIMS) was used to analyze the sphericity, gradient angularity and micro texture. Both volumetric and mechanical properties were studied to evaluate the influence of geometric characteristics of BOF slag coarse aggregate on asphalt concrete. The experimental results show that the BOF slag coarse aggregate has the characteristics of high sphericity, good angular performance and rough surface texture. The geometric characteristics of BOF slag has obvious influence on the volume performance of asphalt concrete. the higher sphericity of BOF slag causes an increase of the air voids of asphalt mixture. BOF slag coarse aggregate can effectively improve the road performances of asphalt concrete. BOF slag’s higher sphericity and angularity improve the moisture damage resistance and rutting resistance of asphalt concrete. Results indicate that better angularity can slightly enhance the moisture resistance property of asphalt concrete, but excessively high angularity of BOF slag coarse aggregates reduces the anti-rutting properties of asphalt mixture.

## 1. Introduction

With the increasing development of the transportation industry in the past few decades, the consumption of natural aggregate resource has been a close concern of scholars in this field. By the end of 2017, the total mileage of expressway has reached 0.137 million kilometers in China [1]. More than 90% of this is asphalt pavement, which has the advantages of low vibration, low noise, short construction time and convenient maintenance [2,3]. Expressway need lots of construction and maintenance, and more than 90% of the components of asphalt pavement are aggregates, which causes the depletion of natural resources [4,5]. Therefore, recycling solid waste and industrial smelting waste are becoming an effective option to relieve the supply pressure of natural aggregate resources [6,7,8].

Steel slag is the main solid waste in steel industry, accounting for more than 10% of crude steel production [9,10,11]. According to the different steelmaking processes, steel slag can be classified into three types: Electric arc furnace (EAF) slag, basic oxygen furnace (BOF) slag and ladle furnace (BF) slag. The large capacity of steel slag can meet the needs of aggregates for road construction, and it has good mechanical properties and high alkalinity [12,13]. The steel slag asphalt mixture has become an increasingly popular research topic in the field of environmental protection road materials in recent years [14,15].

Ahmedzade et al. [16] discussed the application of BOF slag coarse aggregate in AC-10 and AC-5 asphalt mixture. The research showed that the use of BOF slag coarse aggregate in asphalt mixture has better performance than that of limestone. The mechanical properties of steel slag improve the moisture damage resistance and rutting resistance performance of asphalt mixture. Behnood et al. [17,18] used steel slag coarse aggregate in stone mastic asphalt (SMA) mixture and found that steel slag applied in the stone mastic asphalt mixture is feasible. Using steel slag as coarse aggregate in SMA can effectively improve its water damage resistance and pavement durability. Pasetto et al. [19,20] evaluated high performance asphalt concrete with electric arc furnace steel slags. Their results showed that asphalt mixtures with EAF slags exhibited better mechanical characteristics than those of the asphalt mixtures with natural aggregates.

Wu et al. [21,22] studied the properties of BOF slag asphalt mixture from the viewpoint of physicochemical characteristics of basic oxygen furnace (BOF) slag. SEM, XRD, EPMA and other testing methods were used to analyze the physicochemical properties of BOF slag. The results showed that BOF slag has a rough surfaced micro texture, which is beneficial to the bonding properties of asphalt. The BOF slag has good angle properties and a rougher alkaline surface than natural aggregates. The asphalt mixture prepared by the BOF slag has better water damage resistance, and higher temperature rutting resistance, fatigue resistance and anti-skid resistance.

Although existing research has determined that the special physical properties of BOF slag can improve the performance of asphalt concrete, the geometric properties of BOF slag and its influence on asphalt mixture have not been effectively quantified. Nodes et al. [23] analyzed the performance of asphalt pavement with the effect of angularity on natural aggregate. The results showed that higher angularity of aggregates improved the interlocking interaction of coarse aggregates, which contributed to increased rutting resistance of asphalt pavement.

Digital image processing technology can solve this problem properly, as it has been used for accurately quantifying the sphericity, gradient angularity and micro texture of aggregates [24,25,26]. The aggregate image measurement system (AIMS) is a computer automation system that can accurately quantify the geometric properties of aggregates [27]. It uses a high resolution digital camera to quickly picture large numbers of aggregates, and then uses data processing software to calculate and process the shape and surface texture of each aggregate. Finally, quantitative properties, such as the angularity, sphericity and texture of the aggregates, can be obtained [28,29].

This research uses AIMS to quantify and analyze angularity, sphericity and texture of the BOF slag coarse aggregates, and then investigate the influence of geometric characteristics of coarse aggregates on asphalt concrete. The concept of effective density is used in this research to decrease the experimental error caused by the high density of BOF slag. Coarse aggregate of basalt, limestone and three types of BOF slag are used to prepare AC 13 (asphalt mixture with a maximum particle size of aggregate of 13 mm). Volumetric and mechanical properties are discussed to investigate the influence of geometric characteristics of BOF slag coarse aggregate on AC 13.

## 2. Materials and Methods

### 2.1. Aggregates

Two types of natural aggregates, basalt and limestone, were obtained from Hubei province in China for use in this research. Three types of BOF slag were used in this research: BOF slag #1 was obtained from Wuhan Iron and Steel Company in Hubei province, China; BOF slags #2 and #3 were obtained from Baotou Iron and Steel Company in Inner Mongolia. BOF slags 1 and 2 were basic oxygen furnace slag, while BOF slag #3 was pyrolytic BOF slag (basic oxygen furnace slag treated with hot stuffing process). Table 1 lists the chemical compositions based on X-ray fluorescence analysis.

Basalt and limestone are naturally available aggregates. As Table 1 shows, the chemical composition of SiO_2_ in basalt is 47.9%, while chemical composition of CaO in limestone is more than 50%. Compared with naturally available aggregates, the chemical composition of Fe_2_O_3_ in the three types of BOF slag is more than 20%. The SiO_2_ content in the three types of BOF slag is less than 30%.

The basic engineering properties of aggregates were tested according to ASTM standards [30,31,32] and the results are showed in Table 2. They all meet requirements.

Figure 1 shows the diversity between two types of naturally aggregates and three types of BOF slags in SEM images. According to the micrographs of basalt and limestone, the microscopic texture of basalt is relatively dense because it is a volcanic rock, while the microscopic surface of limestone consists of a clastic texture and tiny grain structure as it belongs to the carbonate sedimentary rocks. Comparing micrographs of BOF slags and natural aggregates, BOF slags show different surface textures, particularly on the size and number of surface microscopic pore structures. Micrographs of BOF slags #1 and #2 show that large numbers of pore structures with sizes of approximately 10–50 μm appear on the BOF slag surface. Micrographs of BOF slag #3 shows that the surface microscopic imaging of pyrolytic BOF slag is between basalt and BOF slag, which has a considerable number of pore structures of 1–10 μm in size. The greater the number of surface microscopic pore structures in BOF slag represents higher water absorption and asphalt absorption.

### 2.2. Asphalt Binder and Filler

Asphalt binder was used in this research and its optimum asphalt content is 4.7% in the mixture. Its characteristics are shown in Table 3. Table 4 presents the features of the used limestone filler in this research.

## 3. Experimental Details

### 3.1. Gravity Characteristics of Coarse Aggregate

Aggregate gravity is the first factor to be considered in asphalt mixture design. It includes apparent specific gravity, bulk specific gravity and water absorption.

The specific gravity of BOF slag and natural aggregates is quite different. The purpose of this research is to design an asphalt mixture with different types of coarse aggregates and fine aggregates. Insufficient consideration of aggregate specific gravity will lead to adverse influence on the volumetric and road performance of the asphalt mixture [33].

#### 3.1.1. Aggregate Specific Gravity

Aggregate specific gravity is the ratio of the aggregate’s density to water density at 23 °C. It is quite different between BOF slag and natural aggregate. The apparent specific gravity (Gsa) measures the volume of aggregate particle and impervious voids. The bulk specific gravity (Gsb) measures the volume of aggregate particle, impervious voids and water permeable voids. Water absorption refers to the amount of water absorbed by the void of the aggregate.

Aggregate with different grain size ranges have different aggregate specific gravity. Table 5 shows the Gsb and Gsa of aggregates at different grain size ranges according to the JTG42-2005 standard of China. Table 5 shows that the Gsb and Gsa of BOF slag is higher than that of basalt and limestone. Among three types of BOF slags, the Gsb and Gsa of BOF slag #1 and BOF slag #2 are similar. Due to the self-slaking process of BOF slag by pyrolytic and processing technology, BOF slag #3 (pyrolytic BOF slag) has the highest specific gravity. It is at least 15% higher than the basalt’s specific gravity in each grain size. In the same types of aggregates, the Gsb increase with the increase of aggregate particle size range.

Figure 2 shows the water absorption of the five types of aggregates at different grain size ranges. Immersion duration of aggregates is 24 h. Apparently, with the decrease of aggregate grain size, the water absorption becomes larger. Smaller grain size is logically related to bigger specific surface area, which will therefore decrease the ration of closed voids in aggregates and increase the ratio of open voids. Basalt and limestone have similar water absorption at different grain size ranges. Meanwhile, BOF slag has higher water absorption than natural aggregate. The order of their water absorption from large to small is: #1BOF slag, #2BOF slag and #3BOF slag.

#### 3.1.2. Effective Specific Density

The mix ratio of asphalt mixture with single type of aggregate was based on aggregate weight. However, in the experimental design of this research, four types of aggregates were used to replace basalt coarse aggregate. The aggregate specific gravity results illustrate that big differences appear on specific gravities and water absorption between different aggregate types. Due to the difference in specific gravity, the gradation and optimum asphalt content in the design of asphalt mixture will change.

In the asphalt mixture, the asphalt can partly fill open voids on the surface of aggregate; therefore, the apparent density or bulk density of aggregates cannot effectively represent the density of aggregates in asphalt. It is more effective to use the specific density (Dse), taking into account the volume effect between aggregate types for designing asphalt mixture. The Dse of the different aggregates are shown in Table 6. Coarse aggregate was replaced by the ratio of effective specific density of different aggregates, aiming to reduce the inaccuracy of the test caused by aggregate density.

### 3.2. Preparation of Asphalt Mixture

AC 13 asphalt mixture was selected to investigate the combination of two natural aggregates and three BOF slags. In order to reduce experimental errors, the aggregate gradation was kept constant. The aggregates distribution of basalt was used in the basic aggregate gradation, as shown in Table 7.

The coarse aggregate and fine aggregate play different roles in the asphalt mixture. Coarse aggregates play a major role in supporting and reinforcing the construction of asphalt mixture, while fine aggregates are mainly used to fill the gaps between coarse aggregates. For the AC 13 asphalt mixture, the dividing sieve size between coarse aggregate and fine aggregate is 2.36 mm.

A coarse–fine composition method was proposed and used to keep fine aggregates such as basalt unchanged, while the coarse aggregates were fully replaced basing on effective specific density. Figure 3 explains the detail of coarse–fine composition.

In asphalt concrete, the geometric characteristics of coarse aggregate, such as angularity, sphericity, surface texture, have a great influence on its performance. A marshall stability test was used to characterize the water resistance of asphalt mixture in 60 °C water; this test is still currently used in the road laboratories for the mechanical characterization of asphalt concretes. In order to change geometric characteristics of the coarse aggregate, a ball mill is used to process the coarse aggregate, which imitates the Los Angeles abrasion test. The time of the specific processing was 30 min, at a weight of 3000 g for each sample; the rotating speed was 120 rpm; and using abrasion with 8 steel balls (500 g each). In this research, #1 BOF slag was treated using the above method and named as processed #1BOF slag (SP1). AIMS (aggregate image measurement system) was used to analyze the angularity, sphericity and surface texture of each aggregate.

In this research, the coarse–fine composition of asphalt mixture method was applied, with 0% of coarse aggregates and 100% of fine aggregates passing through sieves of 2.36 mm. Table 8 lists the labels for every coarse–fine composition of asphalt mixture. Four specimens were tested in each and the average data were used for analysis.

### 3.3. Experimental Methods

#### 3.3.1. Aggregates Morphology Test

Shape, angularity, and surface texture of aggregates have been shown to directly affect the engineering properties of highway construction such as HMA (hot mix asphalt concrete) and concrete. The aggregate image measurement system (AIMS) was used to analyze the angularity, sphericity and surface texture of coarse aggregates of each types of aggregate.

As shown in Figure 4, the aggregate image measurement system is comprised of image acquisition hardware and a computer to run the system and process data. The image acquisition hardware used a high-resolution digital camera and a variable magnification microscope to collect digital images and measure aggregates. The system used a tray to place the aggregate under the view of the camera. The first scan used backlighting to describe a profile image of the aggregate particle, from which dimensions and angularity gradients of the edges were measured. The second scan utilized top-lighting and variable magnification to capture texture images and measure each aggregate particle’s height. AIMS then used the computer to process the acquired images. 

Angularity is a description of edge sharpness of the boundary particles of aggregate. The angularity changes with aggregate boundary shape changes. The value of angularity is calculated based on the gradient of the particle boundary. Angularity is calculated with Equation (1) and its range is from 0 to 1000. The lager the value of angularity, the sharper the boundary shape of the aggregate.
(1)$$Angularity=\frac{1}{\frac{n}{3}-1}\overset{n-3}{\underset{i=1}{\sum }}\left|\theta_{i}-\theta_{i+3}\right|$$
where *θ* is angle of orientation of the edge points, *n* is the total number of points, *i* is the ith point on the edge of the particle.

Sphericity is used to characterize the degree of similarity between the aggregate shape and the ideal sphere. Sphericity is calculated with Equation (2), its range is 0 to 1. If the value of sphericity is closer to 1, the shape of aggregate is more similar to the ideal sphere.
(2)$$Sphericity=\sqrt[{3}]{\frac{d_{S}d_{I}}{d_{L}^{2}}}$$
where, *d_S_* is particle shortest dimension, *d_I_* is particle intermediate dimension, *d_L_* is particle longest dimension.

#### 3.3.2. Marshall Stability Test

The Marshall stability test is used to characterize the water resistance of asphalt mixture in 60 °C water. The compacted specimens were divided into two groups according to the numerical average of VV (percent air voids in asphalt mixtures). The first group was put in the water at 60 °C for 30 min. The second group was placed in water at 60 °C for 48 h. The specimens were then placed in a testing machine at a constant displacement load rate of 50 mm/min. We recorded the Marshall stability (MS, maximum load) and Marshall flow (the maximum load starts to decrease when the deformation is mm).

The retained Marshall stability (RMS) is defined as the ratio of the sample to Marshall stability after immersion in hot water for 48 h (MS1, kN) and 30 min (MS, kN). The larger the RMS, the better the water resistance performance of asphalt mixture.

#### 3.3.3. Freeze-Thaw Splitting Test

The freeze-thaw splitting test can be used to measure the ability of moisture resistance of asphalt mixture at low temperatures. The main evaluation index is the splitting strength ratio TSR (%). The larger the value of TSR, the better the moisture resistance of asphalt mixture of the water at freezing and thawing environmental conditions.

Splitting strength ratio (TSR) is defined as the ratio between ITS after F-T circles and ITS of nor-freeze. Among them, the environmental conditions of F-T circles are freezing at −20 °C for 16 h.

#### 3.3.4. Rutting Test

The rutting test was used to characterize the high temperature rutting resistance of asphalt mixture. The size of the rut specimen was 300 × 300 × 50 mm, and the specimen density was controlled at 100 + 1.0% of the Marshall density. The rutting experiment temperature was 60 °C, the wheel pressure was 0.7 MPa, and the experimental wheel round-trip speed was 42 times/min. The dynamic stability (DS) is expressed by the rutting depth change rate of 45–60 min.

### 3.4. Research Program

Figure 5 illustrates the research program on the influence of geometric characteristics of BOF slag coarse aggregate on asphalt concrete. Firstly, the aggregate image measurement system (AIMS) was used to analyze the angularity, sphericity and surface texture of coarse aggregates. Then, the effective specific density was conducted during the asphalt mixture design. Basalt was used to create the AC 13 mixture, and the coarse aggregate was replaced by limestone and BOF slags by coarse-fine composition method, respectively. Volumetric and mechanical properties were studied to evaluate the influence of geometric characteristics of BOF slag coarse aggregate on asphalt concrete.

## 4. Results and Discussion

### 4.1. Geometric Characteristics of Coarse Aggregate

Coarse aggregates with sizes of 9.5–13.2 mm and 4.75–9.5 mm in total, account for 51.3% by the weight of the asphalt mixture. Therefore, the geometric characteristics of coarse aggregate were represented by the angularity, sphericity and surface texture of aggregates with those two sizes. Each test selected 200 aggregate samples, and the illustration of angularity, sphericity and texture is shown in Figure 6.

#### 4.1.1. Angularity

Angularity is a description of edge sharpness of the boundary particles of aggregate. Profile images of different types of aggregate particle from the first scan of AIMS are shown in Figure 7.

The angularity of investigated coarse aggregates is shown in Table 9. This was obtained from the average values of 200 aggregate samples. The results show that the angularity of BOF slag is higher than that of natural aggregates. Different types of BOF slag have different angularity. Among them, in the same place of production, the angularity of basic oxygen furnace slag is higher than pyrolytic BOF slag. Meanwhile, limestone has the lowest angularity and #2 BOF slag has the highest angularity.

#### 4.1.2. Sphericity

The sphericity of the investigated coarse aggregates is shown in Table 10. The results were calculated from the 200 aggregate samples. It can be observed from Table 10 that the limestone had the lowest sphericity and #2 BOF slag had the highest sphericity. BOF slag had a higher degree of sphericity than natural aggregates.

#### 4.1.3. Texture

Texture represents the relative roughness and smoothness of aggregate surfaces. The second scan utilized top-lighting and variable magnification to capture texture images of aggregates; these are shown in Figure 8.

The texture value of an ideal smooth surface is defined as zero. The AIMS texture analysis uses the wavelet method in the calculation and analysis of texture.

The texture of investigated coarse aggregate is shown in Table 11. From the texture values of six types of aggregate, the highest texture value is basalt. The texture value of BOF slag is smaller. As the surface of BOF slag has formed uniform color carbonized layers during the aging process, the photos taken by the digital camera of AIMS cannot represent the surface texture of the BOF slag. With reference to the water absorption of each aggregates in Table 8, #1 BOF slag is considered to be the most texture.

### 4.2. Volumetric Properties of BOF Slag Coarse Aggregate Asphalt Concrete

In traditional asphalt mixture gradation designs the air voids in the asphalt mixture are the most important index of volumetric properties, which can also affect the performance of asphalt concrete. The skeleton structure of AC 13 asphalt mixture is a suspension-compact structure with 3–5% percent air voids in the asphalt mixture.

The percentage of air voids in the asphalt mixture (VV) is defined as the volume percentage of air voids in compacted asphalt mixtures. It is calculated with Equation (2).
(3)$$\mathrm{VV}=100\times \left[1-\left(\frac{G_{f}}{G_{t}}\right)\right]$$
where $G_{t}$ is the theoretical maximum specific gravity of bituminous mixture and $G_{f}$ is bulk specific gravity of bituminous mixture.

Table 12 shows the volumetric properties of AC 13. Compare with group BB, the theoretical maximum density decrease and porosity decrease by about 40% in terms of limestone aggregate. The VV of BS2 is similar with BB, and the use of other groups of BOF slag can effectively improve the value of VV; the highest VV of BSP1 reached 5.83%.

When combining the volumetric properties analysis with geometric characteristics of aggregate, four types of BOF slags have higher sphericity than basalt. Among them, BSP1 has the highest sphericity and the lowest angularity and it has the largest value of VV. BS2 has the highest angularity when compared with BS1, but its VV is lower. This shows that the excessively high angularity reduces the value of VV.

The geometric characteristics of coarse aggregates have a great effect on the volumetric properties of asphalt mixture. Firstly, the rough surface texture and high water absorption of BOF slag coarse aggregate can absorb more free asphalt in the mixture and increase air voids of the asphalt mixture. Secondly, the high sphericity of the BOF slag contributes more skeleton support in the asphalt mix skeleton structure. In addition, BOF slag has a higher angularity than natural aggregate, so the larger sphericity of the BOF slag leads to a higher VV of the asphalt mixture. Lastly, if BOF slag has excessive angularity value, the edge angle of BOF slag would be destroyed during the compaction of the asphalt mixture, which would result in the decrease of the air voids of the asphalt mixture. 

Using BOF slag coarse aggregates will result in a significant increase in the percentage of air voids in asphalt mixture. The rough surface texture and higher sphericity of BOF slag will cause the increase of the air voids of asphalt mixture. Besides, if the BOF slag has excessively high angularity, it will have an adverse effect on the increase of air voids.

### 4.3. Mechanical Properties of BOF Slag Coarse Aggregate Asphalt Concrete

The Marshall stability test, freeze-thaw (F-T) splitting test and rutting test were employed to evaluate the influence of geometric characteristics of BOF slag coarse aggregate on asphalt concrete.

#### 4.3.1. Marshall Stability Test Results

As shown in Figure 9, the values of Marshall stability are presented. The RMS of the basalt-based asphalt mixture has the lowest value of 80.37%. Coarse aggregate of limestone and coarse aggregate of BOF slag can significantly enhance the moisture resistance property. BS1 has the highest water absorption and non-outstanding value of angularity and sphericity. The results of BS1 and BSP1 are similar, showing that improving the sphericity of BOF slag and reducing the angularity of BOF slag have little effect on the moisture resistance property of asphalt mixture.

As can be seen in Figure 10, there are close parabolic equation curve relationships between angularity or sphericity of BOF slag and Marshall stability test results. The R2 of angularity and sphericity is 0.99 and 0.78, respectively. Both of the correlations of angularity and sphericity tend to increase first and then decrease. The optimal angularity and sphericity of Marshall results is 3267 and 0.755, respectively.

AC 13 has a suspension-compactness skeleton structure. The rough surface texture (high water absorption) of BOF slag and good adhesion with asphalt result in a good moisture resistance property of the asphalt mixture. Changing the degree of sphericity or angularity of BOF slag has little influence on the beneficial effect of moisture resistance property on the asphalt mixture.

#### 4.3.2. Freeze-Thaw Splitting Test Results

Figure 11 presents the splitting rest results of composite asphalt mixtures. The tendency of freeze-thaw splitting test is similar to the Marshall stability test. The TSR of BB is the lowest. All types of BOF slags coarse aggregates composite asphalt mixtures possess higher TSR values. There is little difference between the TSR values of BS1 and BSP1. It shows that improving the sphericity of BOF slag and reducing the angularity of BOF slag have little effect on the moisture resistance property of BOF slag coarse aggregate asphalt mixture. TSR value of BS2 is the lowest among the 4 types of asphalt mixture with BOF slag.

As can be seen in Figure 12, there are close parabolic equation curves relationships between angularity of BOF slag and splitting test results, while R2 of angularity is 0.999. The correlations of angularity and TSR tend to increase first and then decrease. The optimal angularity of Marshall results is 3344. There is a weak linear relationship between sphericity and splitting test results.

BOF slag coarse aggregates have a rough surface and an alkaline surface, which enhances the moisture resistance under F-T circles of AC 13. Changing the degree of sphericity of BOF slag has little influence on the improving effect of low temperature moisture resistance property of asphalt mixture. The excessively high angularity of the aggregate can adversely affect the low temperature moisture resistance property of the asphalt mixture.

#### 4.3.3. Rutting Test Results

The Rutting test results are shown in Table 13. It can be clearly observed that the limestone coarse aggregate asphalt mixture has the lowest dynamic stability. Using four types of BOF slag coarse aggregates can improve the values of asphalt mixture dynamic stability in varying degrees, among which BS3 has the best high temperature rutting resistance of asphalt mixture. When compared with BS1, the dynamic stability of BSP1 and BS2 is decreased. This shows that the high sphericity and low angularity of BOF slag coarse aggregates will increase the dynamic stability of the asphalt mixture. Meanwhile, if the BOF slag has excessively high angularity, it will have an adverse effect on the increased dynamic stability.

As can be seen in Figure 13, there are close parabolic equation curves relationships between angularity of BOF slag and dynamic stability, while R2 of angularity is 1.0. The curves of angularity tend to increase first and then decrease. The optimal angularity of dynamic stability is 3285. There is a weak linear relationship between sphericity and dynamic stability. With the rise of sphericity of BOF slag, dynamic stability has a slight reduction trend.

## 5. Conclusions

This study investigated the geometric characteristics (sphericity, angularity and texture) of three types of BOF slag coarse aggregates by AIMS. A coarse-fine composition method and effective density were proposed and studied. Volumetric and mechanical properties of AC 13 were discussed to investigate the influence of geometric characteristics of BOF slag coarse aggregate. The following conclusions were obtained:

(1)Compared with natural aggregates, the microscopic surface of BOF slag has a pitted and vesicular texture and a porous structure with particle range of 1–50 μm. BOF slag coarse aggregates have higher sphericity and angularity than natural aggregates. The BOF slag coarse aggregate has higher water absorption and higher effective density, among which the pyrolytic BOF slag has the largest effective density.(2)Using BOF slag coarse aggregates result in a significant increase on the VV of AC 13. The rough surface texture (higher water absorption) and higher sphericity of BOF slag causes the increase of the air voids in the asphalt mixture. The excessively high angularity of BOF slag can decrease the VV of AC 13 and the higher sphericity of BOF slag causes the increase of the air voids in the asphalt mixture.(3)According to the result of the Marshall stability, freeze-thaw splitting, and Rutting tests, using the coarse aggregate of three types of BOF slag and the processed BOF slag can enhance the properties of moisture damage resistance and rutting resistance of the asphalt mixture. A higher sphericity of BOF slag coarse aggregates leads to better rutting resistance. Excessively high angularity of BOF slag coarse aggregates reduces the anti-rutting properties of asphalt mixture. The optimal angularity of BOF slag is about 3300 based on the analysis of the result of Marshall stability test, freeze-thaw (F-T) splitting test and rutting test.(4)In the production of BOF slag coarse aggregates, the exorbitant angularity of coarse aggregates of BOF slag should be controlled. This will help to improve the service life of BOF slag asphalt pavement and the promotion of slag.

## Figures and Tables

**Figure 1 materials-12-00741-f001:**
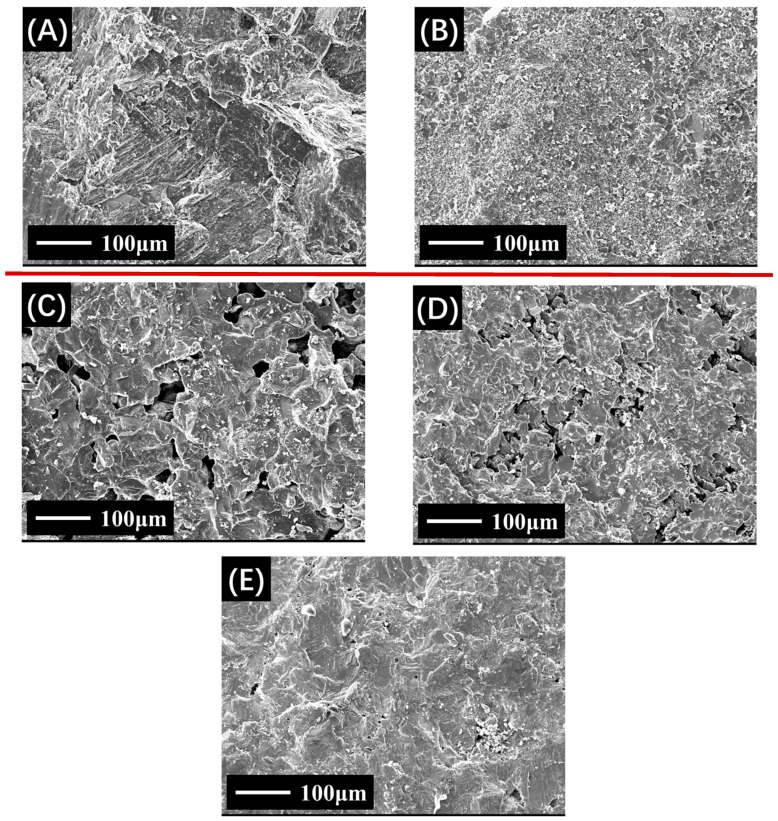
SEM images of investigated aggregates. **A**: Basalt; **B**: Limestone; **C**: #1 BOF slag, **D**: #2 BOF slag, **E**: #3 BOF slag).

**Figure 2 materials-12-00741-f002:**
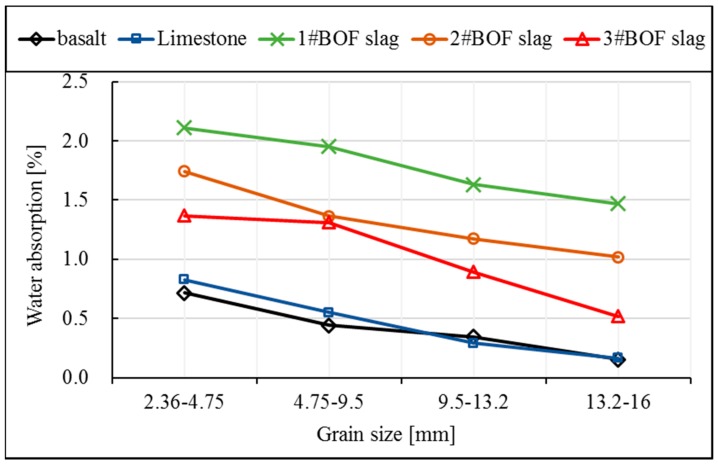
Water absorptions of aggregates.

**Figure 3 materials-12-00741-f003:**
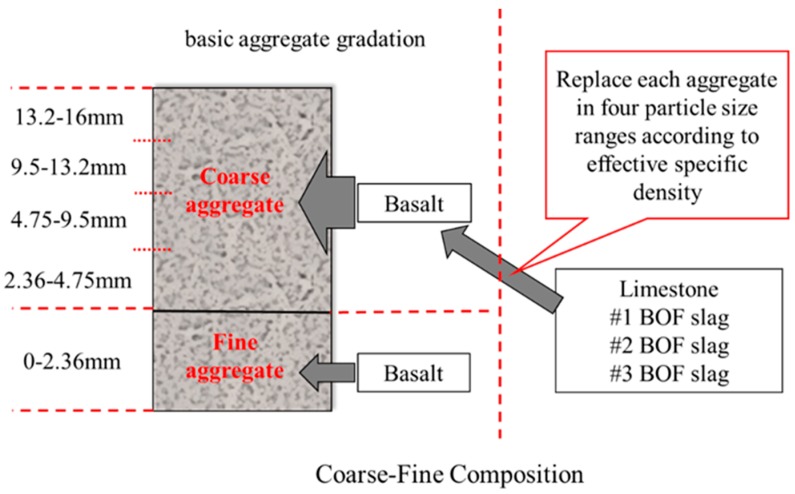
The coarse–fine composition method.

**Figure 4 materials-12-00741-f004:**
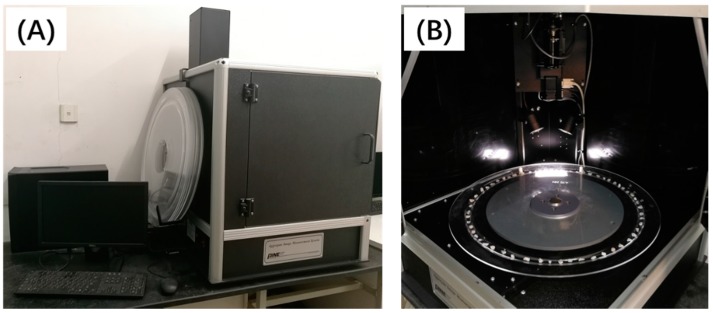
Aggregate image measurement system (**A**), and the tested coarse aggregate placed on tray (**B**).

**Figure 5 materials-12-00741-f005:**
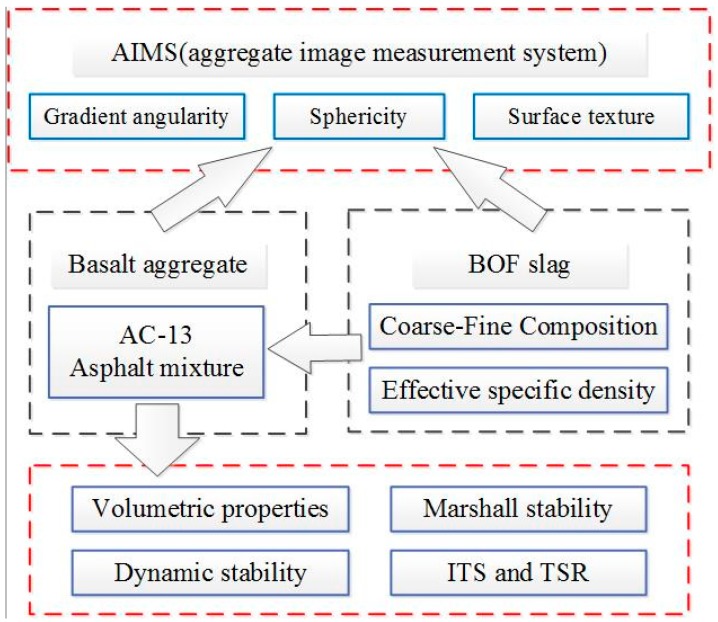
Finalized research program.

**Figure 6 materials-12-00741-f006:**
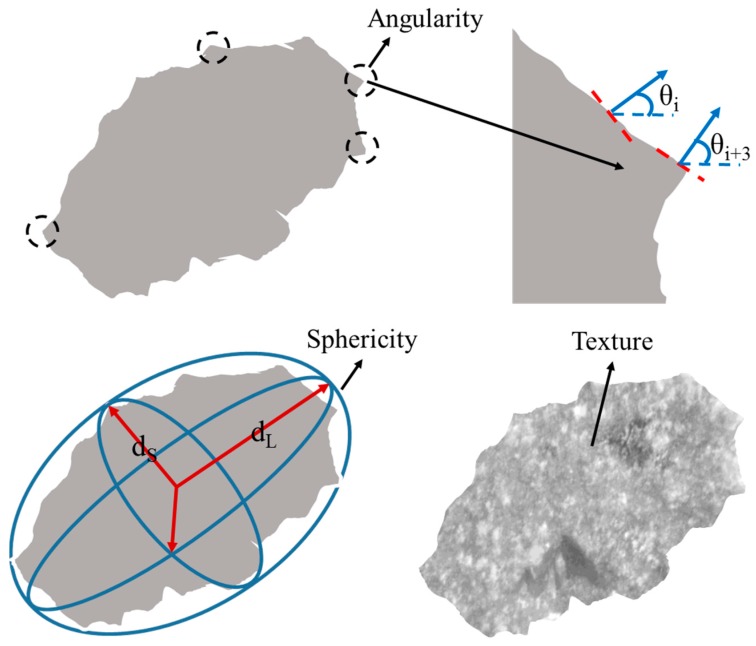
Illustration of angularity, sphericity and texture.

**Figure 7 materials-12-00741-f007:**
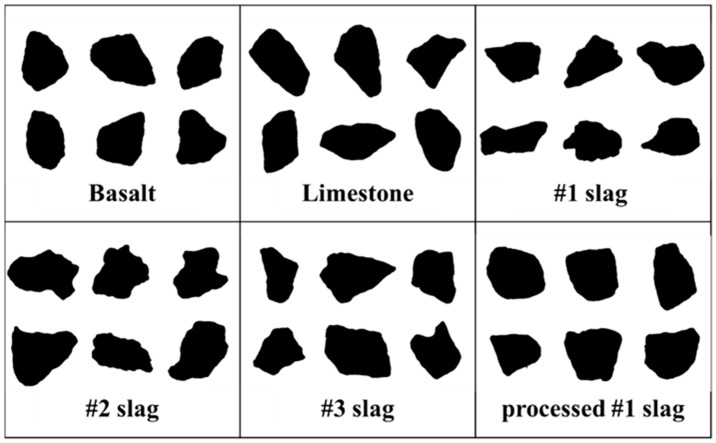
Parts of the profile image of different types aggregate particles.

**Figure 8 materials-12-00741-f008:**
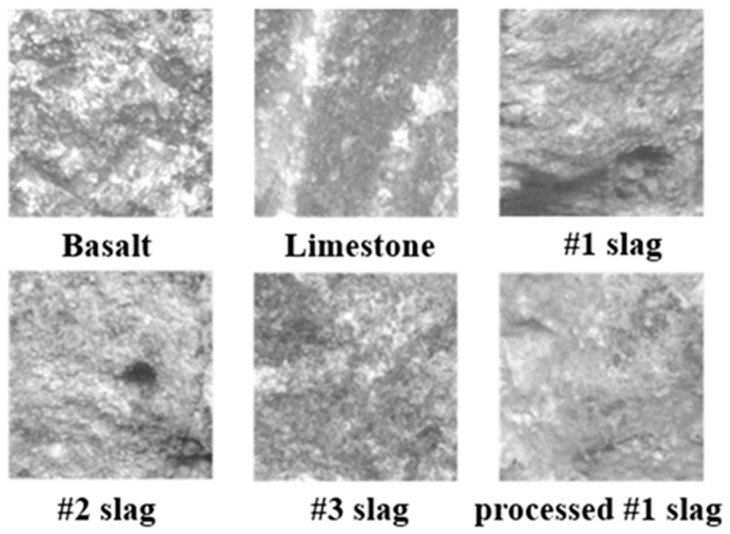
Parts of the surface texture images of aggregates.

**Figure 9 materials-12-00741-f009:**
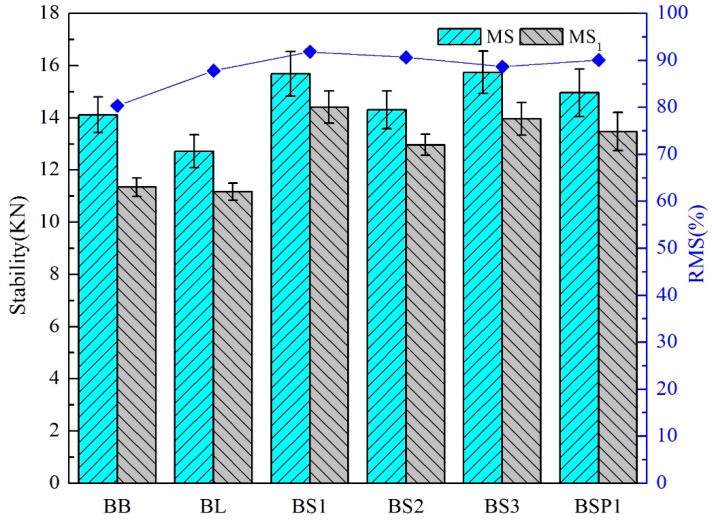
Marshall stability test results with different coarse aggregates.

**Figure 10 materials-12-00741-f010:**
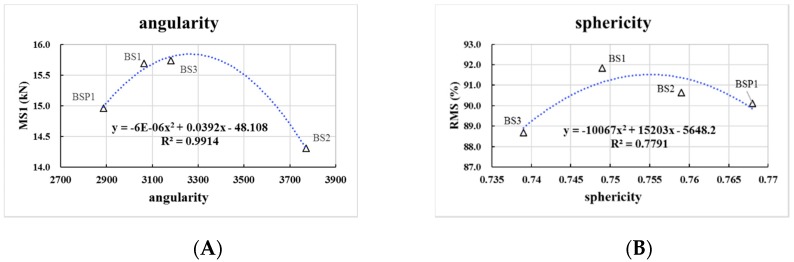
Correlation between Marshall stability test results and aggregate geometric characteristics. (**A**) correlation between MS1 and angularity; (**B**) correlation between RMS and sphericity.

**Figure 11 materials-12-00741-f011:**
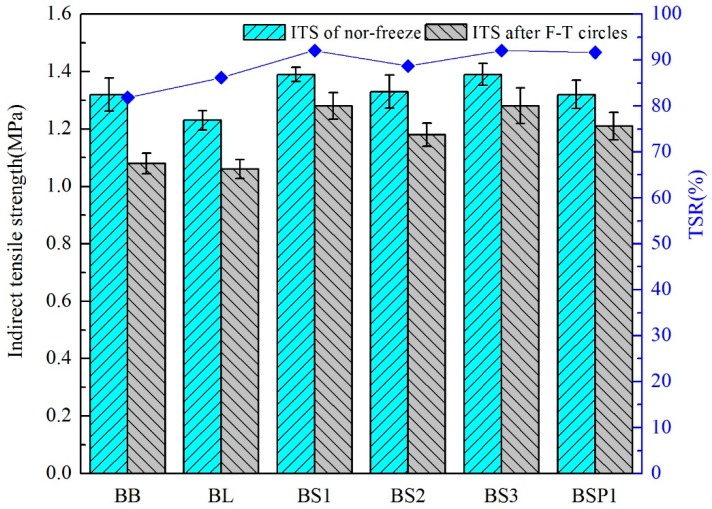
Splitting test results with different coarse aggregates.

**Figure 12 materials-12-00741-f012:**
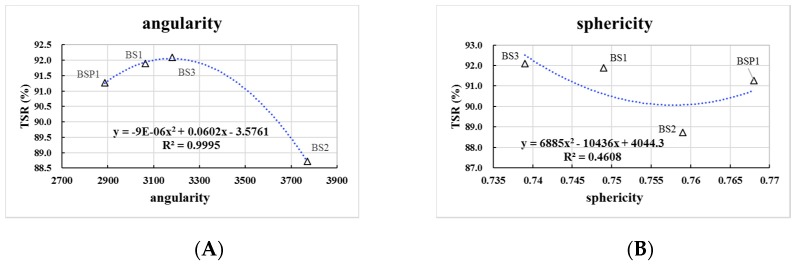
Correlation between splitting test results and aggregate geometric characteristics. (**A**) correlation between TSR and angularity; (**B**) correlation between TSR and sphericity.

**Figure 13 materials-12-00741-f013:**
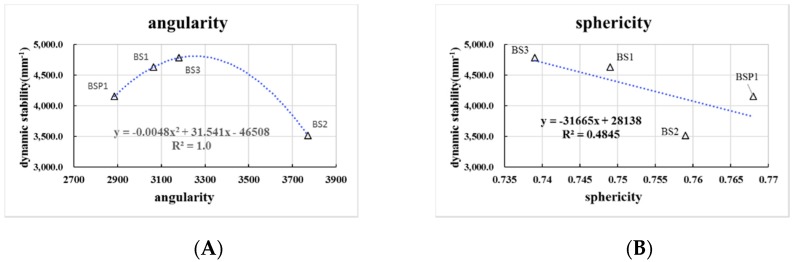
Correlation between rutting test results and aggregate geometric characteristics. (**A**) correlation between dynamic stability and angularity; (**B**) correlation between dynamic stability and sphericity.

**Table 1 materials-12-00741-t001:** Chemical composition of the aggregates.

Aggregate Type	Composition (%)							
	SiO_2_	CaO	MgO	Al_2_O_3_	Fe_2_O_3_	MnO	P_2_O_5_	LOI
Basalt	47.9	8.23	4.34	18.3	9.82	3.6	2.25	5.27
Limestone	0.86	51.2	2.36	0.85	0.12	0.7	1.02	42.7
1# BOF slag	19.2	42.7	5.19	3.25	23.9	1.77	1.41	2.36
2# BOF slag	17.7	39.7	5.56	2.91	24.4	4.55	1.68	3.41
3# BOF slag	15.4	34.4	6.22	1.95	30.8	4.46	2.15	4.46

**Table 2 materials-12-00741-t002:** The basic engineering properties of the aggregates in this research.

Parameter Measured	Basalt	Limestone	BOF Slag			Requirements
			#1	#2	#3	
Los Angeles abrasion (%)	10.2	11.8	16.5	15.6	15.6	≤28
Aggregate Crushing Value (%)	9.1	14.1	14.3	15.2	13.9	≤26
Flakiness and elongation (%)	13.2	10.7	6.6	5.8	4.3	≤18
Fine aggregate angularity (%)	42	46.3	NA	NA	NA	≥30
Sand equivalent (%)	68.5	74.2	NA	NA	NA	≥60

**Table 3 materials-12-00741-t003:** Basic properties of the asphalt binder.

Properties		Values	Specifications
Penetration (0.1 mm)		63	60–80
Penetration index		−0.7	−1.5–1.0
Softening point (°C)		47.7	≥46
Ductility, 5 cm/min, 15 °C (cm]		>160	≥100
Dynamic viscosity (60 °C) (Pa·s)		179	≥160
Density (g/cm^3^)		1.023	
After RTFOT (rotating thin film over test) ageing	Weight loss (%)	−0.05	−0.8–0.8
	Penetration ratio (%)	67.5	≥61
	Residual ductility 5 cm/min, 10 °C (cm)	11.5	≥6

**Table 4 materials-12-00741-t004:** Basic properties of limestone filler in this research.

Properties		Values	Specifications
Apparent specific gravity		2.816	≥2.5
Sieves passing percentage (%)	<0.6 mm	100	100
	<0.15 mm	98.4	90–100
	<0.075 mm	92.2	75–100
Hydrophilic coefficient		0.4	<1.0

**Table 5 materials-12-00741-t005:** Specific gravities of coarse aggregates at different size ranges.

Aggregate	Gravity	Grain Size (mm)			
		2.36–4.75	4.75–9.5	9.5–13.2	13.2–16.0
Basalt	Gsa	2.923	2.918	2.915	2.912
	Gsb	2.863	2.881	2.886	2.899
Limestone	Gsa	2.723	2.716	2.705	2.701
	Gsb	2.663	2.676	2.684	2.689
1#BOF slag	Gsa	3.308	3.302	3.293	3.306
	Gsb	3.092	3.102	3.125	3.153
2#BOF slag	Gsa	3.275	3.263	3.302	3.293
	Gsb	3.098	3.124	3.179	3.186
3#BOF slag	Gsa	3.543	3.587	3.563	3.582
	Gsb	3.379	3.426	3.453	3.517

**Table 6 materials-12-00741-t006:** Dse of aggregates at different grain size range.

Aggregate	Grain Size (mm)			
	2.36–4.75	4.75–9.5	9.5–13.2	13.2–16
Basalt	2.691	2.694	2.695	2.696
Limestone	2.896	2.905	2.902	2.904
#1BOF slag	3.272	3.264	3.217	3.186
#2BOF slag	3.214	3.197	3.245	3.153
#3BOF slag	3.474	3.513	3.498	3.421

**Table 7 materials-12-00741-t007:** Aggregate composite gradation of AC 13 asphalt mixture.

Coarse Aggregates (mm)		Weight Ratio (%)	Fine Aggregates (mm)		Weight Ratio (%)
9.5–16	13.2–16	4.1	0–2.36	1.18–2.36	6.3
	9.5–13.2	22.6		0.6–1.18	7.7
2.36–9.5	4.75–9.5	28.7		0.3–0.6	4.6
	2.36–4.75	13.3		0–0.3	8.7
Filler					4%

**Table 8 materials-12-00741-t008:** Labels for every coarse–fine composition of asphalt mixture.

Fine Aggregate	Coarse Aggregate	Label
Basalt	Basalt (B)	BB
	Limestone (L)	BL
	#1BOF slag (S1)	BS1
	#2BOF slag (S2)	BS2
	#3BOF slag (S3)	BS3
	processed #1BOF slag (SP1)	BSP1

**Table 9 materials-12-00741-t009:** Angularity of investigated coarse aggregates.

Grain Size (mm)	Basalt	Limestone	S1	S2	S3	SP1
9.5–13.2 mm	2965	2896	3064	3770	3180	2886
4.75–9.5 mm	2846	2733	3051	3283	2829	2765

**Table 10 materials-12-00741-t010:** Sphericity of investigated coarse aggregate.

Grain Size (mm)	Basalt	Limestone	S1	S2	S3	SP1
9.5–13.2 mm	0.706	0.662	0.749	0.759	0.739	0.768
4.75–9.5 mm	0.639	0.609	0.693	0.718	0.726	0.729

**Table 11 materials-12-00741-t011:** Texture of investigated coarse aggregate.

Grain Size (mm)	Basalt	Limestone	S1	S2	S3	SP1
9.5–13.2 mm	708	450	349	356	407	419

**Table 12 materials-12-00741-t012:** Volumetric properties with different coarse aggregates.

Aggregate Categories	Theoretical Maximum Density [g/cm^3^]	Apparent Density [g/cm^3^]	VV [%]
BB	2.689	2.585	3.873
BL	2.601	2.545	2.154
BS1	2.828	2.708	4.262
BS2	2.812	2.706	3.769
BS3	2.958	2.831	4.285
BSP1	2.827	2.662	5.832

**Table 13 materials-12-00741-t013:** Rutting text results with different coarse aggregates.

Aggregate Categories	Rutting Depth (mm)	Dynamic Stability (mm)
BB	2.162	3056
BL	3.284	1863
BS1	1.952	4632
BS2	2.341	3517
BS3	1.842	4782
BSP1	2.080	4152

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
