# Peer review of "Geometric Characteristics of BOF Slag Coarse Aggregate and its Influence on Asphalt Concrete"

_materials, 2019, doi:10.3390/ma12050741_

Round 1
Reviewer 1 Report
Tha manuscript proposes the results of an experimental work intended at investigating the role of some relevant geometric properties of recycled aggregates (taken from steel slags) on the resulting performance of asphalt concrete.
The subject is certainly on interest and current relevance, but, in this reviewer's opinion, the style of the presentation is too "empiric" in tone: in fact, no quantitative analysis and comparison with predictive model is proposed.
Moreover, several comments are reported throughout the marked manuscript attached at the present review report: they need to be duly taken into consideration by the authors as they are going to prepare the revised version of their original submission.

Author Response
Point 1: The manuscript proposes the results of an experimental work intended at investigating the role of some relevant geometric properties of recycled aggregates (taken from steel slags) on the resulting performance of asphalt concrete.
Response 1: Thanks for your helpful comments.
Point 2: The subject is certainly on interest and current relevance, but, in this reviewer's opinion, the style of the presentation is too "empiric" in tone: in fact, no quantitative analysis and comparison with predictive model is proposed.
Response 2: Thank you for your comments. We fully agree with you that the style of the presentation is too "empiric" in tone. So we decided to add Figure 10, Figure 12, Figure 13 to analysis correlation between aggregate geometric characteristics and mechanical properties of BOF slag coarse aggregate asphalt concrete. At the same time, the corresponding data analysis is added and highlighted as blue text. One of the analysis as follow.
As can be seen in Figure 10, there are close parabolic equation curves relationships between angularity or sphericity of BOF slag and Marshall results. R2 of angularity and sphericity is 0.99 and 0.78. both the correlations of angularity and sphericity tend to increase first and then decrease. The optimal angularity and sphericity of Marshall results is 3267 and 0.755.
The three types of analysis of correlation between angularity of BOF slag and mechanical properties of asphalt concrete conclusion are similar. The optimal angularity of BOF slag is about 3300.
Point 3: Moreover, several comments are reported throughout the marked manuscript attached at the present review report: they need to be duly taken into consideration by the authors as they are going to prepare the revised version of their original submission.
Response 3: Thank you for your comments. We are very grateful for your kindly careful comments on my paper. We fully agree with your comments reported throughout the marked manuscript. We have mainly discussed your comments and made detailed revisions with highlight as blue text.
We have modified the Funding and Acknowledgements in our paper.
We have added the following references and highlighted as blue text.
2. Skaf M , Manso J M , Aragón, ángel, et al. EAF slag in asphalt mixes: A brief review of its possible re-use. Resources, Conservation and Recycling 2017, 120:176-185.
6. Kambole C , Paige-Green P , Kupolati W K , et al. Basic oxygen furnace slag for road pavements: A review of material characteristics and performance for effective utilisation in southern Africa. Construction and Building Materials, 2017, 148:618-631.
7. Jiang, Y.; Ling, T. C.; Shi, C.; Pan, S. Y., Characteristics of steel slags and their use in cement and concrete—A review. Resources Conservation & Recycling 2018, 136, 187-197.

Reviewer 2 Report
The research topic is quite interesting for the pavement engineering scientific area.
The scientic approach is sound overall, even if one of the mechanical test used, namely the Marshal stability test, is somehow obsolete and really controversial. It is suggested to the authors to modify the text between line 225 and 226 as follows: "Marshall stability test is used to characterize the water resistance of asphalt mixture in 60°C water; such test is still currently used in the road laboratories for the mechanical characterization of asphalt concretes."
At the end of such text it is also suggested to the authors to add the following references:
*) Pasetto, M., Baldo, N. (2006), Electric arc furnace steel slags in "high performance" asphalt mixes: A laboratory characterisation, 2006 TMS Fall Extraction and Processing Division: Sohn International Symposium, Volume 5, 2006, Pages 443-450
*) Y. Xue, S. Wu, H. Hou, J. Zha, Experimental investigation of basic oxygen furnace slag used as aggregate in asphalt mixture, J. Hazard. Mater. B138 (2006) 261-268.
Author Response
Point 1: The research topic is quite interesting for the pavement engineering scientific area.
Response 1: Thanks for your helpful comments.
Point 2: The scientic approach is sound overall, even if one of the mechanical test used, namely the Marshal stability test, is somehow obsolete and really controversial. It is suggested to the authors to modify the text between line 225 and 226 as follows: "Marshall stability test is used to characterize the water resistance of asphalt mixture in 60°C water; such test is still currently used in the road laboratories for the mechanical characterization of asphalt concretes."
Response 2: Thank you for your comments and we fully agree with you. We have modified the text between line 225 and 226 as follows: Marshall stability test is used to characterize the water resistance of asphalt mixture in 60°C water; such test is still currently used in the road laboratories for the mechanical characterization of asphalt concretes.
Point 3: At the end of such text it is also suggested to the authors to add the following references:
*) Pasetto, M., Baldo, N. (2006), Electric arc furnace steel slags in "high performance" asphalt mixes: A laboratory characterisation, 2006 TMS Fall Extraction and Processing Division: Sohn International Symposium, Volume 5, 2006, Pages 443-450
*) Y. Xue, S. Wu, H. Hou, J. Zha, Experimental investigation of basic oxygen furnace slag used as aggregate in asphalt mixture, J. Hazard. Mater. B138 (2006) 261-268.
Response 3: Thank you for your comments and we fully agree with you. We have added the following references in our paper. We also add the research of Pasetto M et al about steel slag in the Introduction of our paper and highlighted as blue text.
14. Ahmedzade, P.; Sengoz, B., Evaluation of steel slag coarse aggregate in hot mix asphalt concrete. Journal of Hazardous Materials 2009, 165, (1), 300-305.
22. Xue Y , Wu S , Hou H , et al. Experimental investigation of basic oxygen furnace slag used as aggregate in asphalt mixture. Journal of Hazardous Materials 2006, 138(2):261-268.

Round 2
Reviewer 1 Report
The Authors have taken in due consideration the commnets raised on their original submission and, hence, the current version of the manuscript can be accepted for publication.